

# On the importance of multiphase photolysis of organic nitrates on their global atmospheric removal

Juan Miguel González-Sánchez[1,2], Nicolas Brun[1,2], Junteng Wu[1], Sylvain Ravier[1], Jean-Louis Clément[2], Anne Monod[1]

[1]Aix Marseille Univ, CNRS, LCE, Marseille, France
[2]Aix Marseille Univ, CNRS, ICR, Marseille, France

*Correspondence to*: Juan Miguel González-Sánchez (juan.gonzalez@mio.osupytheas.fr) and Anne Monod (anne.monod@univ-amu.fr)

**Abstract.** Organic nitrates ($RONO_2$) are secondary compounds, and their fate is related to the transport and removal of $NO_x$ in the atmosphere. While previous research studies have focused on the reactivity of these molecules in the gas phase, their reactivity in condensed phases remains poorly explored despite their ubiquitous presence in submicron aerosol. This work investigated for the first time the aqueous-phase photolysis rate constants and quantum yields of four $RONO_2$ (isopropyl nitrate, isobutyl nitrate, α-nitrooxyacetone, and 1-nitrooxy-2-propanol). Our results showed much lower photolysis rate constants for

these $RONO_2$ in the aqueous phase than in the gas phase. From alkyl nitrates to polyfunctional $RONO_2$, no significant increase of their aqueous-phase photolysis rate constants was observed, even for $RONO_2$ with conjugated carbonyl groups, in contrast with the corresponding gas-phase photolysis reactions. Using these new results, extrapolated to other alkyl and polyfunctional $RONO_2$, as well as other atmospheric sinks (hydrolysis, gas phase photolysis, aqueous and gas phase ·OH oxidation, dry and wet deposition) multiphase atmospheric lifetimes were calculated for 45 atmospherically relevant $RONO_2$ along with the

relative importance of each sink. Their lifetimes range from a few minutes to several hours depending on the $RONO_2$ chemical structure and its water solubility. In general, multiphase atmospheric lifetimes are lengthened when $RONO_2$ partition to the aqueous phase, especially for conjugated carbonyl nitrates for which lifetimes can increase up to 100%. Furthermore, our results show that aqueous-phase ·OH oxidation is a major sink for water-soluble $RONO_2$ ($K_H > 10^5$ M atm$^{-1}$) ranging from 50 to 70 % of their total sink at high LWC (0.35 g m$^{-3}$). These results highlight the importance of investigating the aqueous-phase

$RONO_2$ reactivity to understand how it affects their ability to transport air pollution.

## 1 Introduction

Organic nitrates play a key role in the formation, transport, and removal of $NO_x$. They are secondary compounds formed *via* $NO_x$ + VOC reactions. Depending on their structure, their lifetimes can be long (from a couple of hours to several days) thus they can be transported from polluted areas where they are formed to more remote areas (Shepson, 1999). During their long-

range transport, these molecules are subject to reactions (i.e.: gas-phase photolysis and/or ·OH oxidation) releasing back $NO_x$.



$RONO_2$ are thus responsible for a flatter distribution of $NO_x$ and consequently, they impact other major pollutants such as $O_3$ and secondary organic aerosol (SOA) (Perring et al., 2013). In parallel, $RONO_2$ can remove $NO_x$ from the atmosphere by deposition to the Earth's surface or by transformation into a more inert chemical compound such as nitric acid (Hu et al., 2011; Nguyen et al., 2015). Therefore, their atmospheric reactivity and fate must be considered to accurately predict the transport of

pollution on a regional scale. Besides, this is of special importance for world regions with decreasing $NO_x$ levels (such as Europe and North America) where the $RONO_2$ relative importance in $NO_x$ transport and removal is increasing due to an increase in the overall transformation of $NO_x$ to $RONO_2$ (Romer Present et al., 2020 ).

Numerous studies have investigated the gas-phase reactivity of individual $RONO_2$ molecules (Suarez-Bertoa et al., 2012; Picquet-Varrault et al., 2020; Bedjanian et al., 2018; Talukdar et al., 1997a, b; Clemitshaw et al., 1997; Atkinson and

Aschmann, 1989; Morin et al., 2016), mainly focusing on ·OH-oxidation and photolysis. Their results show that the kinetics and mechanisms of these reactions are highly influenced by each $RONO_2$ chemical structure. Although the presence of the nitrate group in the molecule hinders the ·OH attack, ·OH-oxidation generally represents the main $RONO_2$ gas-phase sink (Shepson, 1999). However, $RONO_2$ with conjugated carbonyl groups are consumed faster *via* photolysis due to an enhancement in their light absorption. This is of high importance for ubiquitous biogenic $RONO_2$ such as isoprene and terpene

$RONO_2$ which often bear conjugated carbonyl groups (Müller et al., 2014; Shen et al., 2021).

$RONO_2$ are not only present in the gas phase, as some of them are low volatile compounds and thus partition into condensed phases. As a result, they represent a fraction ranging from 5 to 77 % of the submicron organic aerosol (Kiendler-Scharr et al., 2016; Ng et al., 2017). Under dry conditions, $RONO_2$ are dissolved in the aerosol phase where the matrix is mostly organic. With increasing relative humidity (RH), the particle can be covered by a water layer where $RONO_2$ (whether it comes from

the particle or the gas phase) can partition. In this case, we consider that soluble $RONO_2$ partitions to the aqueous phase where it exhibits a specific reactivity.

The aqueous-phase reactivity of $RONO_2$ plays a significant role in their atmospheric fate. The hydrolysis of tertiary and allylic $RONO_2$ represents a fast and permanent sink of $NO_x$ in the atmosphere (Hu et al., 2011; Darer et al., 2011; Rindelaub et al., 2015). However, only a small fraction (between 9 % and 34 % for α- and β-pinene $RONO_2$) of the total pool of organic nitrates

undergoes hydrolysis (Takeuchi and Ng, 2019; Wang et al., 2021). Aqueous-phase ·OH-oxidation has been reported to be an important sink for non-volatile terpene $RONO_2$, even though the ·OH attack is more effectively hindered in the aqueous phase than in the gas phase (González-Sánchez et al., 2021). Nevertheless, limited information is available for $RONO_2$ aqueous-phase photolysis. To our knowledge, only one study investigated the aqueous-phase absorption cross-sections of individual $RONO_2$ (Romonosky et al., 2015). However, the literature shows no aqueous-phase photolysis quantum yields, and thus no

$RONO_2$ aqueous-phase photolysis rate constants have been determined so far. Furthermore, the photolysis of $RONO_2$ with conjugated carbonyl groups remains unexplored, despite its high atmospheric relevance.

The objective of this work was to determine the aqueous-phase photolysis rate constants of individual $RONO_2$ including a carbonyl nitrate. Experimental photolysis rate constants of four organic nitrates (isopropyl nitrate, isobutyl nitrate, α-nitrooxyacetone, and 1-nitrooxy-2-propanol) were determined in an aqueous-phase photoreactor, and their average aqueous-




phase quantum yields were determined. Then, atmospheric aqueous-phase photolysis rate constants were calculated under various realistic solar light conditions. Finally, using the experimental results, some estimations were performed for the aqueous-phase photolysis rate constants of a set of 45 atmospherically relevant $RONO_2$, a global evaluation of all atmospheric sinks in both the gas and the aqueous phase was done, and their atmospheric lifetimes were estimated.

## 2 Materials and methods

### 2.1 Aqueous-phase photoreactor


The photolysis experiments were performed in a 450 $cm^3$ double-wall Pyrex aqueous-phase photoreactor (Fig. S1, see details in Renard et al., 2013). The reactor was irradiated by an arc light source (LOT Quantum Design) equipped with a 1000 W arc Xe lamp. Irradiation below 290 nm was removed by an ASTM 892 AM1.5 standard filter. A constant distance (18.4 cm) between the lamp and the water surface was carefully maintained in all experiments using 400 mL of aqueous solution. The

Xe arc lamp spectrum is presented in Fig. 1 (black line) and is compared to the solar actinic flux (red line) on the 1st of July 2015 at 40° latitude at ground level. The lamp actinic flux that reaches the photoreactor was measured by an actinometry study with $H_2O_2$ (detailed in Supplementary Information S1).

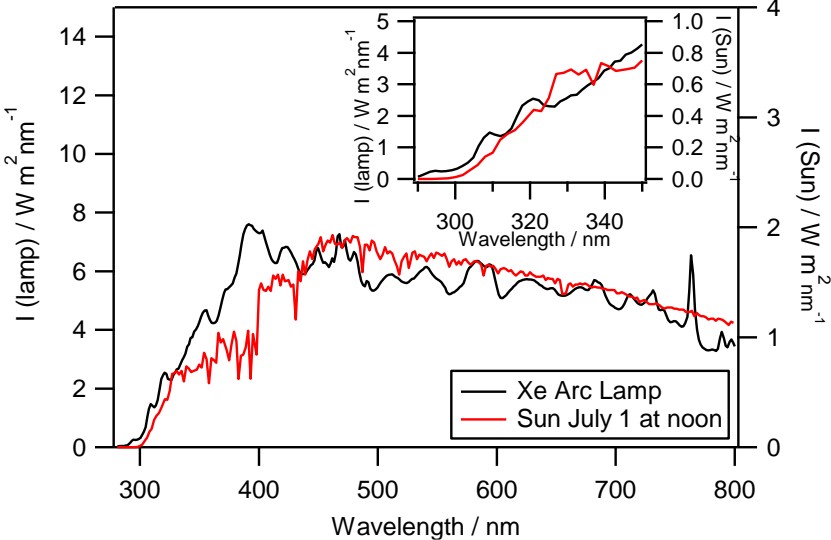

**Figure 1: Irradiation spectra of the Xe 1000 W arc lamp equipped with an AM1.5 filter (black line) compared to the solar irradiation**
**spectra (red line) on the 1st of July 2015 at 40° latitude at ground level with an overhead ozone column of 300 DU and a surface albedo of 0.1 (using the Tropospheric Ultraviolet (TUV) model, Madronich and Flocke, 1999). Inner graph: zoom on the 290 to 350 nm region.**

### 2.2 Experimental protocol and determination of experimental $RONO_2$ photolysis rate constants

Before each photolysis experiment, the photoreactor was filled with 400 mL of ultrapure water and the investigated solubilized

compound. The solution was stirred in the dark for 30 min for complete dissolution of the compound. In parallel, the lamp was



lighted 10 min before the reaction started to stabilize the light beam. Once the reactor was placed underneath the light beam, the first aliquot was sampled signaling the reaction time zero. Photolysis reactions were performed for 7 h at 298.0 ± 0.2 K and "non-controlled" pH (without buffer solutions). During the reaction time, aliquots were sampled regularly (from 2 to 10 minutes) for offline UHPLC-UV analyses. All the performed photolysis experiments, including experimental conditions, and

kinetic results are appended in Table S3. As shown in our previous study (González-Sánchez et al., 2021), isopropyl nitrates and isobutyl nitrate are subject to significant evaporation to the reactor's headspace, and α-nitrooxyacetone is subject to hydrolysis. Therefore, control experiments (under dark conditions) were performed to subtract the evaporation and/or hydrolysis kinetic contributions (see Table S4). Furthermore, slight quantities of ·OH radicals were formed during the photolysis experiments (via photolysis of the produced $HNO_2$ and $NO_2^-$). Since ·OH radicals react with $RONO_2$ (González-

Sánchez et al., 2021), their attack was considered (Eq. (1)) to precisely determine the $RONO_2$ aqueous-phase photolysis rate constant.

$$ln \frac{[RONO_2]_0}{[RONO_2]_t} = k' \cdot t = (J_{RONO_2} + k_{vap/hyd} + k_{OH}[\cdot OH]) \cdot t \ , \qquad (1)$$

where $[RONO_2]_0/[RONO_2]_t$ is the relative decay of aqueous-phase concentrations of the molecule, $k'$ is the pseudo-first-order total decay rate constant ($s^{-1}$), $J_{RONO_2}$ is the experimental photolysis rate constant ($s^{-1}$), $k_{vap}$ and $k_{hyd}$ are the

experimental evaporation and hydrolysis rate constants ($s^{-1}$), respectively, $k_{OH}$ is the aqueous-phase ·OH-oxidation rate constant ($M^{-1} s^{-1}$), $[\cdot OH]$ is the aqueous-phase ·OH radical concentration (M) and $t$ is time (s). A detailed explanation of the estimation of ·OH radical concentration is given in SI (Section S2). Note that the contribution of the ·OH oxidation varies with time since ·OH radicals were secondarily formed. Therefore, only data at the beginning of the reaction (< 2 h) were further employed to minimize this contribution. Under these conditions, aqueous-phase ·OH oxidation of $RONO_2$ accounted for 5 to

10 % of the total decay.

## 2.3 Analyses of aqueous solutions of RONO₂

The absorption cross-sections of $RONO_2$ and $H_2O_2$ in solution were determined from 190 to 340 nm with a UV-Vis-NIR double-beam spectrophotometer (JASCO V670). In addition to the four $RONO_2$ for which experimental photolysis rate constants were investigated (isopropyl nitrate, isobutyl nitrate, α-nitrooxyacetone, and 1-nitrooxy-2-propanol), the absorption

cross-sections of four additional $RONO_2$ were investigated (1-pentyl nitrate, isopentyl nitrate, 2-ethylhexyl nitrate, and isosorbide 5-mononitrate). Due to low absorption cross-sections above 290 nm, a 5 cm path length cell was used, and due to the low water solubility of some $RONO_2$, methanol or methanol/water mixture was used as a solvent. The solvents' effects on the absorption cross-sections were investigated for isopropyl nitrate. Table S5 lists the solvents and ranges of concentrations used to investigate the absorption cross-sections of each $RONO_2$ (and also $H_2O_2$)

The investigated organic nitrates showed an intense UV absorption band around 200 nm (Fig. S2). During the photolysis experiments, isopropyl nitrate, isobutyl nitrate, α-nitrooxyacetone, and 1-nitrooxy-2-propanol were monitored by UHPLC-



UV. The instrument was a Thermo Scientific Accela 600 equipped with a Hypersil Gold C18 column (50 × 2.1mm) with a particle size of 1.9 μm and an injection loop of 5 μL at 200 nm. A binary eluent of $H_2O$ and $CH_3CN$ was used for all analyses at a flow rate of 400 μL min$^{-1}$. Two gradients were used depending on the compounds' polarity. For α-nitrooxyacetone and 1-

nitrooxy-2-propanol, the gradient started from $H_2O/CH_3CN$ 90/10 (v/v) to 50/50 (v/v) for 3 min, held at this proportion for 1 min, and then set back to 90/10 (v/v) within 10 s until the end of the run, at minute 5. For isopropyl nitrate and isobutyl nitrate, a similar gradient was employed but the initial and final proportions were $H_2O/CH_3CN$ 80/20 (v/v).

Calibration curves were optimized to obtain good linearity between $5 \times 10^{-5}$ and $1 \times 10^{-3}$ M with an $R^2 > 0.9995$. The retention times were 0.9, 1.2, 2.4, and 3.33 min for 1-nitrooxy-2-propanol, α-nitrooxyacetone, isopropyl nitrate, and isobutyl nitrate

respectively. Limits of detection were $1 \times 10^{-5}$ M for 1-nitrooxy-2-propanol, and α-nitrooxyacetone, and $9 \times 10^{-6}$ M for isopropyl nitrate and isobutyl nitrate.

**2.4 Reagents**

Chemicals were commercially available and used as supplied: isopropyl nitrate (96%, Sigma Aldrich), isobutyl nitrate (98%, Sigma Aldrich), 2-ethylhexyl nitrate (97%, Sigma Aldrich), 1-pentyl nitrate (98%, TCI Chemicals), isopentyl nitrate (98%,

TCI Chemicals), isosorbide 5-mononitrate (98%, Acros Organics), $H_2O_2$ (30%, non-stabilized, Acros Organics). Non-commercially $RONO_2$, α-nitrooxyacetone, and 1-nitrooxy-2-propanol were synthesized and purified (see SI Section S3). LC/MS grade Acetonitrile (Fisher Optima) was used as supplied. Tap water was purified with a Millipore MiliQ system (18.2 MΩ cm and TOC < 2 ppb).

**3 Results and discussions**

**3.1 Liquid-phase absorption cross-sections of RONO₂**

UV-Vis absorbance was investigated for eight organic nitrates (i.e., isopropyl nitrate, isobutyl nitrate, 1-pentyl nitrate, isopentyl nitrate, 2-ethylhexyl nitrate, α-nitrooxyacetone, 1-nitrooxy-2-propanol, and isosorbide 5-mononitrate) dissolved in water or methanol. All $RONO_2$ absorbed UV light between 190 nm and 330 nm, with maximum absorption at 210 nm due to the π → π* transition (see Fig. S2). At longer wavelengths, the absorption is produced due to an n → π* transition. This

transition appears as a shoulder of the π → π* transition and extends up to ~ 330 nm. The n → π* transition is responsible for the light absorption at λ ≥ 290 nm, and thus, is relevant for the atmospheric photolysis of $RONO_2$. The determination of the liquid-phase absorption cross-sections of $RONO_2$ is detailed in SI (Section S4).

**3.1.1. Liquid-phase absorption cross-sections of alkyl nitrates**

Figure 2 shows the absorption cross-sections determined for all investigated alkyl nitrates and are compared to those reported

in the literature (both in the liquid and the gas phase). Since the absorption cross-sections of alkyl nitrates were mostly investigated in methanol due to their low water solubilities, the absorption cross-sections of isopropyl nitrate in water were





compared to that in methanol. The comparison showed that there is a slight shift to shorter wavelengths (blue shift) when isopropyl nitrate is dissolved in water ($\sim$ 5 nm in the most impacted region). This shift is likely caused by the stabilization of the non-bonding orbital with increasing solvent polarity. Since this stabilization lowers the ground state energy, the n $\rightarrow \pi^*$

transition energy increases, and thus shorter wavelengths are needed to promote the electron.

It can be concluded from Fig. 2 that all the investigated alkyl nitrates showed similar absorption cross-sections in the liquid phase. The absorption cross-sections determined for isopropyl nitrate are in good agreement with those determined by Romonosky et al., (2015). However, for 2-ethylhexyl nitrate, slightly lower values were obtained in this work at $\lambda < 300$ nm, even though a more polar solvent was used in our work, the reason for this difference is not clear.

Compared to the gas-phase absorption cross-sections, isobutyl nitrate and 1-pentyl nitrate present a $\sim$40 % increase in solution. For isopropyl nitrate this increase is less important, the absorption cross-sections are $\sim$ 25% higher in methanol and nearly identical in water.





**Figure 2: Absorption cross-sections of alkyl nitrates in methanol and/or water. Gas-phase absorption cross-sections are included (in red) when available (for isopropyl nitrate, isobutyl nitrate, and 1-pentyl nitrate).**






### 3.1.2. Liquid-phase absorption cross-sections of polyfunctional $RONO_2$

Figure 3 shows the absorption cross-sections determined for the investigated polyfunctional $RONO_2$ dissolved in water or
water/methanol. 1-Nitrooxy-2-propanol and isosorbide 5-mononitrate presented absorption cross-section values similar to the
investigated alkyl nitrates (Fig. 2). In contrast, α-nitrooxyacetone absorption cross-sections were around five times higher (note
the different scale used for this molecule) due to the conjugation of the carbonyl and the nitrooxy group. Furthermore, there
are differences in the shape of its spectra: a large shoulder is observed from 320 to 390 nm. This band is not observed in the
gas phase and thus it might be caused by interactions between the two chemical groups and the solvent, or it could correspond
to an impurity that remained after synthesis, even after purification.

For isosorbide 5-mononitrate absorption cross-sections, the values observed in this work were similar to those determined by
Romonosky et al., (2015), although our values were slightly lower at $\lambda > 310$ nm.

Comparison between liquid- and gas-phase absorption cross-sections show that UV absorption is significantly enhanced when
polyfunctional $RONO_2$ are in solution. For α-nitrooxyacetone, this enhancement is of a factor of 2 compared to the values
determined by Barnes et al., (1993), and of one order of magnitude compared to the values determined by Roberts and Fajer,
(1989).

Figure 3 also shows the absorption cross-sections of several β-hydroxy $RONO_2$ (listed in Table 2) in a liquid phase and one
(2-nitrooxy-1-ethanol) in the gas phase. The absorption cross-sections are higher by an order of magnitude for molecules in
solution. The only other β-hydroxy $RONO_2$ for which gas-phase absorption cross-sections were investigated, i.e., *trans*-2-
nitrooxy-1-cyclopentanol, are not shown here since the molecule did not absorb UV light above 275 nm (Wängberg et al.,
1996). These observations suggest that the nitrate group absorption is likely hindered by the hydroxy group in the gas phase
but not in solution, probably due to solvent effects. Nevertheless, gas-phase absorption cross-sections should be determined
for other hydroxy $RONO_2$ to confirm this hypothesis. This is of special atmospheric relevance since β-hydroxy nitrates are
formed *via* the addition of ·OH radicals to atmospherically relevant unsaturated molecules (such as terpene nitrates and
aromatic nitrates, for example) and may significantly partition between the gas and the condensed phases in the atmosphere.




**Figure 3: Absorption cross-sections of polyfunctional RONO₂ in water or water/methanol determined in this work and in**
**Romonosky et al., (2015). Gas-phase absorption cross-sections of polyfunctional RONO₂ are included when available (for α-**
**nitrooxyacetone, and 2-nitrooxyethanol). β-Hydroxy nitrates A, B, C, D, F, and H are listed in Table 2.**

### 3.2 Liquid-phase photolysis quantum yields of RONO₂

The photolysis quantum yields of RONO₂ were estimated in the liquid phase for the first time. For that purpose, the maximum
theoretical photolysis rate constants of RONO₂ were calculated assuming quantum yields of unity and then compared with the
experimentally determined ones.

The maximum theoretical photolysis rate constants of RONO₂ under our experimental conditions were calculated using Eq.
(3):



$$J_{calc} = \int \sigma(\lambda)\Phi(\lambda)\,I(\lambda)d\lambda, \tag{3}$$

where $J_{calc}$ was the calculated photolysis rate constant (s⁻¹), I(λ) the corrected lamp actinic flux (photons s⁻¹ cm⁻² nm⁻¹), and
Φ(λ) the quantum yield (assumed equal to one). Since the calculated values represented maximum rate constants using liquid-phase quantum yields of unity, Eq. (4) was used to estimate the actual quantum yields, assuming a constant value over 290 – 340 nm.

$$\Phi = \frac{J_{exp}}{J_{calc}}, \tag{4}$$

where $J_{exp}$ is the experimental rate constant (s⁻¹, Table 1).

Quantum yields are given only for isopropyl nitrate, isobutyl nitrate, α-nitrooxyacetone, and 1-nitrooxy-2-propanol (investigated for $J_{exp}$). Furthermore, for each molecule, while quantum yields may vary with λ, an average quantum yield was determined over atmospherically relevant wavelengths (290 to 340 nm). Wavelength-resolved quantum yields might be important for α-nitrooxyacetone since two distinct absorbance bands were observed at the investigated wavelengths (Fig. 3). Additionally, isobutyl nitrate and α-nitrooxyacetone photolysis rate constants might be overestimated due to the solvent used
for the determinations of their absorption cross-sections. Indeed, for isopropyl nitrate, a 60 % increase in the calculated photolysis rate constant was observed when dissolved in methanol as compared to water (Table 1). However, no corrections were performed for isobutyl nitrate and α-nitrooxyacetone since the enhancement reported for isopropyl nitrate is not necessarily extendable to other molecules.

**Table 1. Calculated and experimental photolysis rate constants of RONO2 in the liquid phase, estimated liquid-phase quantum yields (Φ), and comparison with gas-phase Φ.**

| RONO₂ | Solvent for J$_{calc}$ | J$_{calc}$ (x10⁻⁵ s⁻¹) | J$_{exp}$ (x10⁻⁵ s⁻¹) | Liquid-phase Φ | Gas-phase Φ |
|---|---|---|---|---|---|
| Isopropyl nitrate | water | 1.69 ± 0.43 | 0.50 ± 0.10 | 0.29 ± 0.09 | 1.00 ± 0.05[a] |
|  | methanol | 2.86 ± 0.38 |  |  |  |
| Isobutyl nitrate | methanol | 2.43 ± 0.89 | 0.94 ± 0.34 | 0.39 ± 0.20 | 1[c] |
| 1-Pentyl nitrate | methanol | 1.92 ± 0.69 |  |  |  |
| Isopentyl nitrate | methanol | 3.30 ± 0.52 |  |  |  |
| 2-Ethylhexyl nitrate | methanol | 2.31 ± 0.41 |  |  |  |
| Isosorbide 5-mononitrate | water | 2.12 ± 0.20 |  |  |  |
| 1-Nitrooxy-2-propanol | water | 6.16 ± 0.85 | 0.40 ± 0.04 | 0.07 ± 0.01 | 1[c] |
| α-Nitrooxyacetone | water/methanol | 172 ± 16 | 0.31 ± 0.02 | 0.002 ± 0.001 | 0.9[b] |

[a]Experimentally determined at 308, 315, and 320 nm by Carbajo and Orr-Ewing, (2010). [b]Estimated by Müller et al., (2014) out of data from Suarez-Bertoa et al., (2012). [c]Assumed to be similar to alkyl nitrates gas-phase Φ.



Table 1 shows the calculated and experimental photolysis rate constants along with the estimated quantum yields. It clearly shows that the calculated values were similar for all compounds, except for 1-nitrooxy-2-propanol and α-nitrooxyacetone. These compounds presented much higher values due to their stronger UV absorption. In contrast, the aqueous-phase experimental photolysis rate constants were of the same order of magnitude for the four investigated $RONO_2$, and no increase associated with the presence of any functional group adjacent to the nitrate group was observed.

Table 1 also shows that quantum yields are much lower in the liquid phase than in the gas phase. The estimated quantum yields are ~3, ~15, and ~500 times lower in solution for alkyl nitrates, 1-nitrooxy-2-propanol, and α-nitrooxyacetone, respectively. This observation is coherent with previous studies showing that the photolysis quantum yields in the aqueous phase are usually lower than those in the gas phase, as shown, for example, for $H_2O_2$ or $HNO_3$ (Herrmann, 2007; Bianco et al., 2020; Romer et al., 2018). This can be caused by the solvent cage effect. When a molecule is photolyzed in a solvent, its photolysis products

are trapped by the surrounding solvent molecules. This solvent cage eases the reconversion of the products into the original molecule, and thus decreases the overall quantum yield. Additionally, excited molecules can easily lose the gained energy by colliding with the surrounding solvent molecules.

    Although the absorption cross-sections of polyfunctional $RONO_2$ such as α-nitrooxyacetone and 1-nitrooxyacetone are enhanced in solution, the enhancement does not imply an increase in their photolysis rate constants since their quantum yields

are also lower in solution. The same effect has been reported for the photolysis of $NO_3^-$ in aqueous solutions when compared to the gas-phase photolysis of $HNO_3$ (Svoboda et al., 2013; Warneck and Wurzinger, 1988; Nissenson et al., 2010).

    For α-nitrooxyacetone, the extremely low quantum yield determined in this work is influenced by the important absorption band observed above 320 nm (Fig. 3). However, even when removing this band from its absorption cross-sections (by deconvolution of the spectra, see SI Section S5), a very low quantum yield was obtained (0.02), lower than any other $RONO_2$

due to its higher absorption between 290 and 320 nm. In any case, the aqueous phase photolysis rate constant of this compound was similar to the other $RONO_2$ despite its much higher UV absorption. This is of special importance since gas-phase photolysis is one of the major sinks for carbonyl $RONO_2$ (Müller and Stavrakou, 2005). These results indicate that if these compounds effectively partition to the aqueous phase, their photolysis may not be such a relevant sink.

    To evaluate the atmospheric impact of aqueous-phase photolysis, its rate constants were calculated under various light

conditions for the investigated organic nitrates and seven other $RONO_2$ molecules for which liquid-phase absorption cross-sections were reported by Romonosky et al., (2015).

### 3.3 Atmospheric aqueous-phase photolysis rate constants of $RONO_2$

    Atmospheric aqueous-phase photolysis rate constants were calculated for fourteen $RONO_2$ using Eq. (3) under two different scenarios (Table 2): i) a global scenario (actinic flux with a 60° solar zenith angle), and ii) a summer scenario (actinic flux for

the 1$^{st}$ of July at noon at 40° latitude). The latter was investigated to determine the maximum aqueous phase photolysis kinetics of $RONO_2$ under atmospheric conditions. The actinic flux was taken from the Tropospheric Ultraviolet-Visible (TUV) model (Madronich and Flocke, 1999). Other parameters in common for both scenarios were an overhead ozone column of 300 DU,



a surface albedo of 0.1, and a ground elevation of 0 km. Although it has been discussed that light is enhanced in clouds' water droplets by a factor of two (Madronich, 1987), this enhancement factor was not included here since it can largely fluctuate.

Furthermore, the comparison with the gas-phase photolysis rate constants appears clearer if no enhancement factor is included. The investigated $RONO_2$ comprised five alkyl nitrates, one ketonitrate, and eight hydroxy nitrates including seven β-hydroxy nitrates, and four $RONO_2$ conjugating more than one functional group (Table 2). The absorption cross-sections were either calculated in this work or taken from Romonosky et al., (2015) who determined the absorption cross-sections of some $RONO_2$ dissolved in a mixture of water and methanol (50/50, v/v). Using our results shown above (Table 1), a quantum yield of 0.34

(average from isopropyl nitrate and isobutyl nitrate) was applied to all alkyl nitrates and isosorbide 5-mononitrate, a quantum yield of 0.07 was applied for all β-hydroxy nitrates, and a quantum yield of 0.002 was applied for α-nitrooxyacetone. The aqueous-phase photolysis lifetimes were estimated using Eq. (5).

$$\tau_{h\upsilon} = \frac{1}{J_{aq} \cdot 86400}, \tag{5}$$

where $\tau_{h\upsilon}$ is the aqueous-phase photolysis lifetime (in days), and $J_{aq}$ is the aqueous-phase photolysis rate constant (in $s^{-1}$).


**Table 2. Calculated aqueous-phase photolysis rate constants and lifetimes for a series of $RONO_2$ under two scenarios (global and summer), and comparison with their gas-phase values.**

| RONO₂ | Aqueous phase | | | | Gas phase | |
|---|---|---|---|---|---|---|
| | $J_{global}$ $(\times 10^{-7} s^{-1})$ | $\tau_{h\upsilon,global}$ (d) | $J_{summer}$ $(\times 10^{-7} s^{-1})$ | $\tau_{h\upsilon,summer}$ (d) | $J_{global}$ $(\times 10^{-7} s^{-1})$ | $\tau_{h\upsilon,global}$ (d) |
| Isopropyl nitrate | 3.2[a] | 36 | 10 | 12 | | |
| | 4.2[b] | 27 | 14 | 8 | 8.9[c] | 14 |
| Isobutyl nitrate | 5.9[a] | 20 | 17 | 7 | 5.3[d] | 22 |
| 1-Pentyl nitrate | 3.0[a] | 38 | 11 | 11 | 8.9[e] | 13 |
| Isopentyl nitrate | 9.7[a] | 12 | 25 | 5 | | |
| 2-Ethyl hexyl nitrate | 4.4[a] | 26 | 14 | 8 | | |
| | 3.5[b] | 33 | 12 | 10 | | |
| α-Nitrooxyacetone | 0.56[a] | 210 | 1.4 | 77 | 92[f] | 1 |
| Isosorbide 5-mononitrate | 5.6[a] | 21 | 15 | 8 | | |
| | 2.1[b] | 54 | 7.5 | 16 | | |



| | | | | | | |
|---|---|---|---|---|---|---|
| 1-Nitrooxy-2-propanol | 4.2[a] | 27 | 10 | 11 | 0.07[g] | 1593 |
| **A** | 1.8[b] | 66 | 5.1 | 23 | | |
| **B** | 4.2[b] | 27 | 11 | 10 | | |
| **C** | 2.5[b] | 47 | 6.6 | 18 | | |
| **D** | 4.7[b] | 25 | 12 | 10 | | |
| **F** | 9.7[b] | 12 | 24 | 5 | | |
| **H** | 18[b] | 6 | 47 | 2 | | |

**Aqueous-phase absorption cross-sections determined in [a]this work, [b]Romonosky et al., (2015). Average gas-phase absorption cross-sections were taken from [c]Clemitshaw et al., (1997), Roberts and Fajer, (1989), and Talukdar et al., (1997); [d]Clemitshaw et al., (1997) and Roberts and Fajer, (1989); [e]Clemitshaw et al., (1997); and [f]Barnes et al., (1993) and Roberts and Fajer, (1989) [g]Value from Roberts and Fajer, (1989) corresponding to 2-nitrooxyethanol.**

Table 2 shows that all aqueous-phase photolysis rate constants were similar except for α-nitrooxyacetone, and the molecule labeled **H**. For the latter, absorption cross-sections (determined by Romonosky et al., (2015)) were higher than for all the other investigated RONO$_2$ (Fig. 3), probably due to the conjugated ester and vinyl groups in the molecule. Likely, this molecular structure inferred a quantum yield lower than 0.07, and thus its photolysis rate constant was probably overestimated in Table 2. Furthermore, although slight differences were observed between this work and that of Romonosky et al., (2015) for isopropyl nitrate and 2-ethylhexyl nitrate, significant differences were obtained for isosorbide 5-mononitrate. This is due to slightly higher determined absorption cross-sections at 310 to 340 nm (Fig. 3). At these wavelengths, the high intensity of the solar actinic flux provokes a substantial variation in the photolysis rate constants.

The apparent similarities between all the RONO$_2$ aqueous-phase photolysis rate constants are of special importance since it might indicate that RONO$_2$ molecules show very similar aqueous-phase photolysis lifetimes independently of the group the molecules bear in addition to the nitrate function.

Table 2 shows that, in general, aqueous-phase global photolysis lifetimes are quite long (from 6 to 210 d), and thus RONO$_2$ can remain in the aqueous phase for several days if they do not hydrolyze or undergo other sinks. The aqueous-phase photolysis lifetimes are shortened by 2.4 to 3.5 times using a solar spectrum at noon on the 1[st] of July.

The comparison between the aqueous-phase and the gas-phase photolysis lifetimes shows that for alkyl nitrates, no significant changes are appreciated. However, strong deviations are observed for α-nitrooxyacetone and 1-nitrooxy-2-propanol.

α-Nitrooxyacetone presents a much longer photolysis lifetime in the aqueous phase (210 vs. 1 d) due to the extremely low photolysis quantum yield in the aqueous phase (0.002). This implies that the aqueous-phase photolysis is a negligible sink for



α-nitrooxyacetone. As mentioned above, gas-phase photolysis is the major sink for carbonyl nitrates as their kinetics are enhanced by the conjugation of the nitrate and the carbonyl group. Our results show that this enhancement is hindered in the aqueous phase, and thus photolysis might not be the major sink for carbonyl nitrates partitioning to the aqueous phase. Conversely, the compared pair of β-hydroxynitrates (1-nitrooxy-2-propanol and 2-nitrooxyethanol) show a much shorter lifetime in the aqueous phase (16 vs. 1593 d) due to a large increase in the absorption cross-sections (Fig. 3). Therefore, the

photolysis sink of β-hydroxynitrates is likely greatly enhanced if they partition to the atmospheric aqueous phase. Nevertheless, the obtained very long lifetimes suggest that aqueous-phase photolysis remains a negligible sink.

To investigate the relative importance of the aqueous-phase photolysis in the RONO$_2$ atmospheric fate, the multiphase lifetimes of several atmospherically relevant RONO$_2$ were calculated as exposed in the next section.

## 4 Atmospheric implications

### 4.1 Photochemical sink contributions to RONO$_2$ multiphase lifetimes

Multiphase photochemical lifetimes were calculated for 32 non-hydrolyzable atmospherically relevant RONO$_2$ classified in three families according to their VOC precursor and water solubility: six small RONO$_2$ (isopropyl nitrate, isobutyl nitrate, 1-pentyl nitrate, isopentyl nitrate, 1-nitrooxy-2-propanol, and nitrooxyacetic acid) with low to intermediate water solubilities ($K_H \sim 10^{-1}$ to $10^5$ M atm$^{-1}$), five isoprene nitrates (ethanal nitrate, α-nitrooxyacetone, two methyl vinyl ketone nitrate isomers,

and a C$_5$ dihydroxy dinitrate) with intermediate to high water solubilities ($K_H \sim 10^3$ to $10^7$ M atm$^{-1}$), and twelve terpene nitrates (α- and β-pinene, limonene, γ-terpinene, and myrcene atmospheric reactivity products) with intermediate to very high water solubilities ($K_H \sim 10^4$ to $10^{12}$ M atm$^{-1}$). The chemical structures of the investigated RONO$_2$ and their Henry's Law constants are listed in Table S7.

In the same manner as in González-Sánchez et al., (2021), the partitioning between the gas and aqueous phase was investigated

under two different scenarios: i) under cloud/fog conditions (LWC = 0.35 g m$^{-3}$), and ii) under wet aerosol conditions (LWC = 3 · 10$^{-5}$ g m$^{-3}$). The partitioning of each molecule in the aqueous phase was calculated using Eq. (6):

$$\varphi_{aq} = \frac{n_{aq}}{n_{aq}+n_{gas}} = \frac{1}{1+(1/LWC_v \cdot K_H \cdot R \cdot T)}, \tag{6}$$

where $\varphi_{aq}$ is the molar fraction of the compound in the aqueous phase, $n_{aq}$ and $n_{gas}$ are the number of moles of RONO$_2$ in the aqueous and the gas phase, respectively; $K_H$ is the Henry's Law constant at standard conditions (in M atm$^{-1}$); R is the ideal

gas constant (0.082 atm L mol$^{-1}$ K$^{-1}$); T is the temperature (set at 298 K) and $LWC_v$ is the liquid water content in volume units (m$^3$ of water/m$^3$ of air). Experimental $K_H$ values were taken from (Sander, 2015) when available, or they were calculated using the GROHME method (Raventos-Duran et al., 2010).

Multiphase photochemical lifetimes were calculated using Eq. (7):

$$\tau_{multiphase} = \frac{1}{\varphi_{aq} \cdot J_{aq} + \varphi_{aq} \cdot k_{OH,aq} \cdot [OH]_{aq} + \varphi_{gas} \cdot J_{gas} + \varphi_{gas} \cdot k_{OH,gas} \cdot [OH]_{gas}}, \tag{7}$$





where $\varphi_{gas}$ is the molar fraction of the compound in the gas phase; $J_{aq}$ and $J_{gas}$ (s$^{-1}$) are the global photolysis rate constants (actinic flux with a 60° solar zenith angle) in the aqueous and the gas phase, respectively; $k_{OH,aq}$ and $k_{OH,gas}$ (in M$^{-1}$ s$^{-1}$ and cm$^3$ molecules$^{-1}$ s$^{-1}$, respectively) are the aqueous and gas-phase ·OH-oxidation rate constants; and $[OH]_{aq}$ and $[OH]_{gas}$ (in M and molecules cm$^{-3}$, respectively) are the ·OH concentrations in each phase.

Experimental values were used for $k_{OH,aq}$ and $k_{OH,gas}$ when available or they were calculated using the group contribution

methods presented in González-Sánchez et al., (2021) and Jenkin et al., (2018), respectively. $J_{aq}$ and $J_{gas}$ were calculated as described in Section 3 when experimental aqueous or gas-phase absorption cross-sections were available. When not available, average values were employed according to the molecule's chemical structure. For all investigated RONO$_2$, a unique $J_{aq}$ value of $3.9 \cdot 10^{-7}$ s$^{-1}$ was considered. This value is the average of eight RONO$_2$ molecules bearing a carbonyl or a hydroxy group: α-nitrooxyacetone, 1-nitrooxy-2-propanol, isosorbide 5-mononitrate, and compounds A, B, C, D, and F in Table 2. This value

is likely a good approximation since all the RONO$_2$ photolysis rate constants determined in this work fall in the same order of magnitude and do not present significant deviations (Table 2). For RONO$_2$ with undetermined gas-phase absorption cross-sections, three values were used depending on their chemical structure. A high value ($6.6 \cdot 10^{-5}$ s$^{-1}$) was used for carbonyl nitrates since these compounds present an enhancement in their photolysis rates compared to alkyl nitrates. This value was averaged from all carbonyl nitrates photolysis rate constants available in the literature (see Table S8). For the C$_5$ dihydroxy

dinitrate compound, a value of $4.4 \cdot 10^{-6}$ s$^{-1}$ was chosen, averaged from dinitrates photolysis rate constants. For other RONO$_2$, a value of $7.6 \cdot 10^{-7}$ s$^{-1}$ was chosen, averaged from photolysis rate constants of alkyl nitrates with more than 2 carbon atoms. The ·OH concentrations were set to $10^{-14}$ M in the aqueous phase and $1.4 \cdot 10^6$ molecules cm$^{-3}$ in the gas phase (Tilgner et al., 2013).

Figure 4 depicts the RONO$_2$ multiphase lifetimes under both cloud/fog conditions (Fig. 4a) and wet aerosol conditions (Fig.

4b). The chosen RONO$_2$ were distributed into three groups according to their nature and source and plotted by increasing water solubility. Figure 4 also shows the aqueous molar fraction of each molecule (in blue) and the relative contribution of each of the investigated sinks to the total multiphase lifetime.



**Figure 4: Chemical multiphase lifetimes and relative contribution of each sink for 32 atmospherically relevant RONO₂ distributed into i) small RONO₂ (SN, left axis), ii) isoprene nitrates (IN, right axis), and iii) terpene nitrates (TN, right axis) under (a) cloud/fog conditions (LWC = 0.35 g m⁻³) and (b) wet aerosol conditions (LWC = 3 x10⁻⁵ g m⁻³). The numbers in blue indicate the aqueous-phase molar fraction (in %). *Conjugated carbonyl nitrates. The chemical structures, properties, and kinetic rate constants of each compound are listed in Table S7.**



Figure 4 shows much longer multiphase photochemical atmospheric lifetimes for small $RONO_2$ (from 38 to 264 h) than for
isoprene and terpene nitrates (from 2 to 29 h). This is mainly due to their low number of $\cdot$OH attack reactive sites, and the
absence of highly reactive groups such as aldehyde groups (fast $\cdot$OH oxidation) or conjugated carbonyl groups (fast gas-phase
photolysis).

The figure also highlights the relevance of aqueous-phase $\cdot$OH oxidation, which is the only photochemical sink for $RONO_2$
partitioning into the aqueous phase. Photolysis is a negligible sink in the aqueous phase, whereas it is an important sink in the
gas phase, especially for compounds bearing conjugated carbonyl groups (marked with * in the figure).

Figure 4 also shows that $RONO_2$ multiphase photochemical atmospheric lifetimes can substantially vary under different
atmospheric LWC. $RONO_2$ lifetimes generally increase when the compound effectively partitions to the aqueous phase. This
increase is especially important for compounds bearing conjugated carbonyl groups due to the significant difference in their
photolysis kinetics between the gas and the aqueous phase. In the gas phase, photolysis is their major sink, while it becomes a
minor or even negligible sink when they partition to the aqueous phase. Besides, $RONO_2$ with no conjugated carbonyl groups
tend to show a mild increase in their lifetimes when partitioning to the aqueous phase caused by the deactivation of their $\cdot$OH
reactivity in water (González-Sánchez et al., 2021).

Comparing Fig. 4a and 4b, very different behaviors are observed depending on the $RONO_2$ Henry's Law constant. The
lifetimes of $RONO_2$ with low water solubilities ($K_H < 10^4$ M atm$^{-1}$, i.e., alkyl nitrates, 1-nitrooxy-2-propanol, α-
nitrooxyacetone, ethanal nitrate, α-pinene 1 and β-pinene 1), barely vary between the cloud/fog and the wet aerosol scenarios
since their aqueous-phase molar fractions are extremely low ($\varphi_{aq} \leq 7$ %).

In contrast, for molecules with intermediate to high water solubilities ($K_H = 10^5 - 10^9$ M atm$^{-1}$, i.e., methyl vinyl ketone isomers,
$C_5$ dihydroxy dinitrate, α-pinene 2–4, β-pinene 2–8, terpinene 1), significant variations between the two scenarios are clearly
observed. Their aqueous-phase partitioning ranges from 91 % (on average) under cloud/fog conditions to 12 % (on average)
under wet aerosol conditions. The increase of photochemical lifetimes observed under cloud/fog conditions is much more
pronounced for $RONO_2$ bearing conjugated carbonyl groups (lifetimes up to 3 times greater) due to their much slower aqueous
phase reactivity.

Finally, very high water-soluble $RONO_2$ ($K_H \geq 10^{10}$ M atm$^{-1}$, i.e., β-pinene 8–11, limonene 1, and myrcene 1–4) barely partition
to the gas phase even under low LWC ($\varphi_{aq} \geq 89$ % under wet aerosol conditions), and thus, their lifetimes are similar under
both conditions. For these $RONO_2$, aqueous-phase $\cdot$OH oxidation is the main sink, even under conditions with extremely low
amounts of water.

## 4.2 Importance of hydrolysis in multiphase chemical lifetimes

Hydrolysis is a known process that can influence the atmospheric lifetimes of $RONO_2$, mostly tertiary and allylic $RONO_2$.
These chemical structures can stabilize the reaction intermediate carbocation formed through the acid-catalyzed unimolecular
nucleophilic substitution ($S_N1$). Other $RONO_2$ (such as primary or secondary $RONO_2$) can also undergo hydrolysis under very



acidic conditions (Rindelaub et al., 2016; Wang et al., 2021), but these reactions remain extremely slow under atmospheric conditions, and they are considered in this work as "non-hydrolyzable".

Up to date, the hydrolysis of nine tertiary $RONO_2$ and four allylic $RONO_2$ have been experimentally investigated by different authors (Hu et al., 2011; Darer et al., 2011; Jacobs et al., 2014; Rindelaub et al., 2015; Wang et al., 2021), reporting a wide

range of hydrolysis rate constants at neutral pH (from $9.9 \cdot 10^{-6}$ to $9.3 \cdot 10^{-3}$ $s^{-1}$).

To compare the importance of aqueous-phase ·OH-oxidation and photolysis towards hydrolysis, the multiphase lifetimes of these $RONO_2$ were evaluated under the two scenarios (cloud/fog and wet aerosol conditions) using Eq. (8).

$$\tau_{multiphase} = \frac{1}{\varphi_{aq} \cdot k_{hyd} + \varphi_{aq} \cdot J_{aq} + \varphi_{aq} \cdot k_{OH,aq} \cdot [OH]_{aq} + \varphi_{gas} \cdot J_{gas} + \varphi_{gas} \cdot k_{OH,gas} \cdot [OH]_{gas}},$$  (8)

The chemical structures of the investigated compounds are described in Table S9 along with their Henry's Law constants (from

1 to $10^{10}$ M $atm^{-1}$) and their hydrolysis rate constants. Their photolysis and ·OH-oxidation rate constants were assumed or estimated as described in Section 4.1. Nevertheless, eight of the investigated molecules (all the allylic $RONO_2$ and *tert* 1, 2, 5, and 6) bear an unsaturation in their chemical structure, and thus their aqueous-phase ·OH-oxidation rate constant cannot be calculated with the Structure-Activity Relationship (SAR). For these compounds, a rate constant of $10^{10}$ $M^{-1}$ $s^{-1}$ was assumed. This assumption is based on the high reactivities (close to the diffusion limit) of unsaturated molecules due the ·OH addition

on the double bond (Herrmann et al., 2015). Besides, it should be noted that no experimental photolysis rate constants were reported for allylic $RONO_2$ in any phase. Hence, the assumed photolysis rate constants may be inaccurate.

Figure 5 displays the hydrolyzable $RONO_2$ multiphase atmospheric lifetimes in the same manner as in Fig. 4, and it shows that hydrolysis can substantially impact $RONO_2$ atmospheric removal.

On the one hand, for $RONO_2$ with high hydrolysis rate constants (*tert* 3, 4, 7, 8, and 9, and *ally* 3 and 4 with $k_{hyd} > 10^{-3}$ $s^{-1}$)

the hydrolysis is the major sink under cloud/fog conditions ) even for compounds that barely partition to the aqueous phase (*tert* 3 and 4); while under wet aerosol conditions, it can be a very significant sink (*tert* 7, 8 and 9). These $RONO_2$ are processed within less than 2 h under cloud/fog conditions, and their atmospheric lifetimes can be shortened by two orders of magnitude with respect to the wet aerosol scenario. Only the highly water-soluble *tert* 9 presents similar lifetimes (0.7 h) under both conditions. These results show that for $RONO_2$ with high hydrolysis rate constants, aqueous-phase ·OH oxidation and

photolysis are completely irrelevant, and their chemical lifetimes are only driven by hydrolysis or gas-phase reactivity depending on their atmospheric partitioning and hydrolysis rate constants.





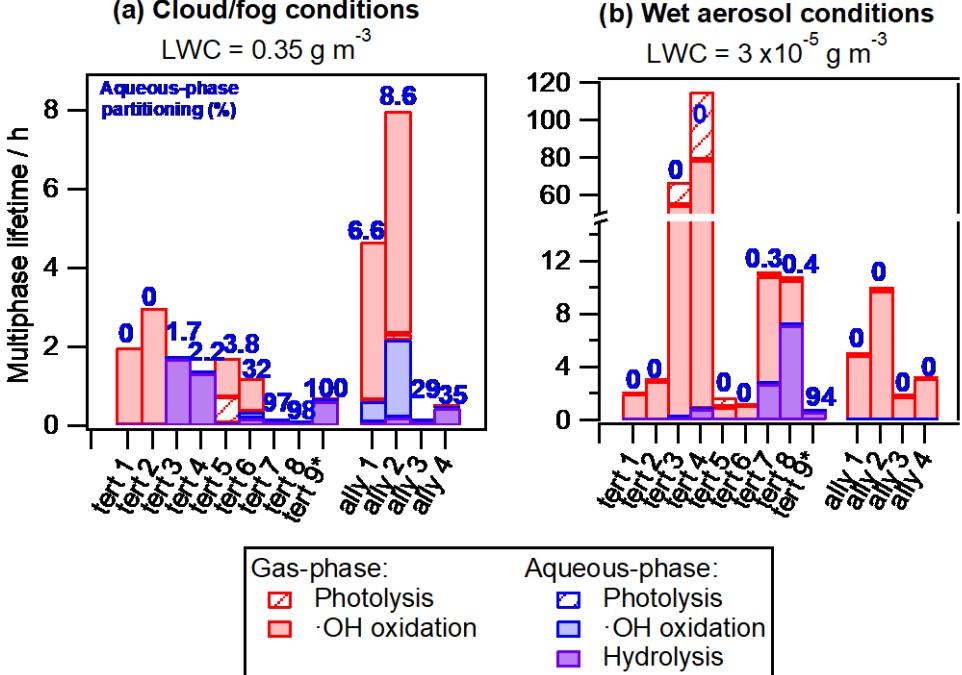

**Figure 5: Chemical multiphase lifetimes and relative contribution of each sink of tertiary and allylic RONO₂ under (a) cloud/fog conditions (LWC = 0.35 g m⁻³) and (b) wet aerosol conditions (LWC = 3 x10⁻⁵ g m⁻³). The numbers in blue indicate the aqueous-phase molar fraction (in %). The chemical structures, properties, and kinetic rate constants of each compound are listed in Table S9.**

On the other hand, for RONO₂ with lower hydrolysis rate constants ($k_{hyd} < 10^{-4}$ s⁻¹), aqueous-phase ·OH oxidation can compete with or overcome hydrolysis as a faster sink (*tert* 6, and *ally* 1 and 2 under cloud/fog conditions). Nevertheless, their atmospheric removal is mostly controlled by their gas-phase reactivity due to their low water solubility. However, it is likely that the aqueous-phase ·OH oxidation is an important process for other tertiary and allylic RONO₂ with higher water solubilities, such as RONO₂ bearing carbonyl groups and/or presenting low hyperconjugation since these molecules tend to present longer hydrolysis lifetimes (Wang et al., 2021). This is of high importance since many authors tend to assume short atmospheric lifetimes due to fast hydrolysis (within hours) for large fractions of atmospheric RONO₂ (Fisher et al., 2016; Zare et al., 2019; Browne et al., 2013), while the decay of these RONO₂ can actually be mostly controlled by the aqueous-phase ·OH reactivity.

## 4.3 Overall RONO₂ multiphase lifetimes

Finally, overall multiphase lifetimes of 45 RONO₂ were calculated by including the contribution of dry and wet deposition using Eq. (9). These 45 compounds were classified into i) non-hydrolyzable small RONO₂, ii) non-hydrolyzable isoprene terpene nitrates, and iii) hydrolyzable RONO₂.





$\tau_{multiphase} = \dfrac{1}{\varphi_{aq} \cdot k_{hyd} + \varphi_{aq} \cdot J_{aq} + \varphi_{aq} \cdot k_{OH,aq} \cdot [OH]_{aq} + \varphi_{gas} \cdot J_{gas} + \varphi_{gas} \cdot k_{OH,gas} \cdot [OH]_{gas} + k_{dep}}$,   (9)

where $k_{dep}$ is the deposition rate constant (in s$^{-1}$) that accounts for both dry and wet deposition.

Dry deposition rate constants were calculated using a deposition velocity of 1.4 cm s$^{-1}$ (averaged from RONO$_2$ deposition velocities determined by Nguyen et al., (2015)) and considering a boundary height layer of 1000 m. It is likely that dry deposition rate constants overestimate the deposition of alkyl nitrates, since the molecules investigated by Nguyen et al., (2015)

are polyfunctional RONO$_2$. Wet deposition rate constants were determined following the work of Brimblecombe and Dawson, (1984) assuming an annual rainfall rate of 1 m yr$^{-1}$ (see Section S6). Since the rainfall rate is assumend yearly wet deposition rate constants are considered equal in both cloud/fog and wet aerosol scenarios.

Figure 6 shows the overall lifetimes of all investigated RONO$_2$ under both cloud/fog conditions (Fig. 6a), and wet aerosol conditions (Fig. 6b). Furthermore, the relative contribution of each sink (deposition, aqueous-phase reactivity, and gas-phase

reactivity) is represented.

The results show that all sinks (deposition, aqueous-phase and gas-phase reactivity) significantly contribute to the RONO$_2$ atmospheric consumption. However, the contribution of each sink and the overall RONO$_2$ atmospheric lifetimes depend largely on the RONO$_2$ chemical structure and the LWC. These parameters can thus highly impact the NO$_x$ atmospheric transport. Hereafter, we discuss the results for each class of RONO$_2$.

**Non-hydrolyzable small RONO$_2$.** Due to their low reactivity, they present the highest lifetimes (between 13 and 19 h). These lifetimes barely vary under both investigated scenarios since the compounds hardly partition to the aqueous phase in any scenario. In our calculations, deposition controls their atmospheric sink (81 % on average). However, dry deposition rate constants are likely overestimated for alkyl nitrates since the molecules investigated by Nguyen et al., (2015) are polyfunctional RONO$_2$. Hence, these RONO$_2$ likely present higher atmospheric lifetimes. Under these circumstances, non-hydrolyzable small

RONO$_2$ are the major responsible for NO$_x$ flatter distribution due to their long-range atmospheric transport.







**Figure 6: Multiphase atmospheric lifetimes and relative contributions of each sink for 45 atmospherically relevant RONO$_2$ distributed into i) non-hydrolyzable small RONO$_2$ (SN), ii) non-hydrolyzable isoprene nitrates (IN) and terpene nitrates (TN), and iii) hydrolyzable RONO$_2$ (HN) under (a) cloud/fog conditions (LWC = 0.35 g m$^{-3}$), and (b) wet aerosol conditions (LWC = 3 x10$^{-5}$ g m$^{-3}$).**





**Non-hydrolyzable isoprene and terpene nitrates.** They present shorter lifetimes (between 3 and 12 h) than those of small $RONO_2$. In general, their atmospheric lifetimes are mostly controlled by chemical sinks although deposition is considerable (43% and 38 % under cloud/fog and wet aerosol conditions, respectively). Their average lifetimes are slightly longer under cloud/fog conditions (5.6 h vs 5.0 h under wet aerosol conditions) due to their lower reactivity in the aqueous phase. This

increase in the atmospheric lifetime with increasing LWC is especially important for $RONO_2$ with intermediate to high water solubility $K_H = 10^5 - 10^9$ M atm$^{-1}$) and the presence of conjugated carbonyl groups (the increase is of up to twice). The less-reactive nature of $RONO_2$ in the aqueous phase increases the relative contribution of deposition sinks (up to twice) under cloud/fog conditions. Therefore, for this kind of compounds, an LWC increase would result in lower $NO_x$ recycling efficiencies since deposition represent a permanent $NO_x$ sink although $RONO_2$ are likely transported further. One should also note the

importance of aqueous-phase reactivity for terpene nitrates even under wet aerosol conditions.

**Hydrolyzable $RONO_2$.** Much shorter lifetimes are estimated under cloud/fog conditions (average 1.6 vs. 5.1 h) mostly due to the fast hydrolysis of $RONO_2$ with high hydrolysis rate constants (*tert* 1 – 5, and *ally* 3 and 4). The atmospheric lifetimes of these $RONO_2$ can be shortened by two orders of magnitude. Due to the shortening on their atmospheric lifetimes and the irreversible loss of the nitrate group through hydrolysis, these $RONO_2$ likely transport much less effectively $NO_x$ at increasing

LWC.

## 5 Conclusions

Photolysis rate constants and quantum yields were determined in the liquid phase for the first time for isopropyl nitrate, isobutyl nitrate, α-nitrooxyacetone, and 1-nitrooxy-2-propanol. Photolysis of these compounds was shown to be hindered in the liquid phase compared to the gas phase. Although they generally presented higher absorption cross-sections when dissolved in the

liquid phase, lower quantum yields were observed compared to the gas phase (0.005–0.29 versus ~1), probably due to solvent cage effects.

Furthermore, no significant differences were observed in the aqueous-phase photolysis rate constants between various $RONO_2$ containing a carbonyl group, a hydroxy group, or none of them. In contrast, previous studies have shown that the gas-phase photolysis of $RONO_2$ is greatly enhanced for α- or β-carbonyl nitrates. Our results showed much lower photolysis rates for

carbonyl nitrates in the aqueous phase and thus, longer photolysis atmospheric lifetimes than in the gas phase. This is of special relevance for these compounds in the atmosphere since photolysis is expected to be their major sink.

Considering two different scenarios: i) cloud/fog conditions (LWC = 0.35 g m$^{-3}$) and ii) wet aerosol conditions (LWC = 3 × 10$^{-5}$ g m$^{-3}$), a complete evaluation of the atmospheric sinks of 45 $RONO_2$ was performed, including aqueous- and gas-phase ·OH-oxidation and photolysis, hydrolysis, and dry and wet deposition. The results highlighted the importance of aqueous-

phase ·OH oxidation, a major sink for some $RONO_2$ even under low LWC, whereas aqueous-phase photolysis remained of negligible importance. The results also emphasized the influence of the $RONO_2$ chemical structure on $RONO_2$ atmospheric



fate and thus their ability to transport $NO_x$. The chemical structure of each $RONO_2$ can influence the kinetics of its multiphase reactivity, and its partitioning between the aqueous and the gas phase.

Small $RONO_2$ such as alkyl nitrates barely partition into the aqueous phase (even under high LWC conditions). Furthermore,
they present low gas-phase reactivity due to their low number of reactive sites and absence of highly reactive groups. Hence, these $RONO_2$ present the longest lifetimes and thus, are responsible for the $NO_x$ flatter distribution.

The atmospheric fate of polyfunctional $RONO_2$ such as isoprene and terpene nitrates highly depends on their chemical structure. Some tertiary and allylic $RONO_2$ present very high hydrolysis rate constants ($k_{hyd} > 5 \cdot 10^{-4}\,s^{-1}$). For these compounds, hydrolysis is the only sink even when they mildly partition to the aqueous phase ($\varphi_{aq} > 0.4\,\%$). Their atmospheric lifetimes
can decrease drastically (up to two orders of magnitude) with increasing LWC, and under these conditions, their processing represents a net sink of $NO_x$. Out of these results, it is evident that more research should be done to clearly elucidate the influence of the $RONO_2$ chemical structure on the hydrolysis rate constant.

The fate of polyfunctional $RONO_2$ with low or negligible hydrolysis ($k_{hyd} < 1 \cdot 10^{-4}\,s^{-1}$) is mainly controlled by their atmospheric partitioning. For molecules with low water solubility ($K_H < 10^4\,M\,atm^{-1}$), their fate is mainly controlled by their
gas-phase reactivity and dry deposition. Their atmospheric lifetimes are lower than those of alkyl $RONO_2$ (ranging from 1 to 10 h) and are impacted by the presence of conjugated carbonyl groups (fast photolysis), and aldehyde groups, and double bonds (fast ·OH-oxidation). They may thus recycle $NO_x$. $RONO_2$ with intermediate water solubilities ($K_H = 10^5 - 10^9\,M\,atm^{-1}$) show more complex processing, their atmospheric fate and lifetimes highly depend on the LWC. At high LWC, their sink is mainly controlled by aqueous-phase ·OH oxidation and dry and wet deposition, while at low LWC gas-phase ·OH oxidation
and photolysis are the main sinks. Due to the decrease of the $RONO_2$ reactivity in condensed phases, their atmospheric lifetimes increase with increasing LWC (up to twice greater). Furthermore, the overall importance of non-chemical sinks (deposition) increases with higher LWC. The ability of $RONO_2$ aqueous-phase ·OH-oxidation to recycle $NO_x$ must be investigated to properly predict the impact of their fate, especially since they represent an important fraction of the atmospherically relevant $RONO_2$. Finally, $RONO_2$ with very high water solubility ($K_H \geq 10^{10}\,M\,atm^{-1}$) partition to the aqueous phase even under very
low LWC, and thus their fate is mainly controlled by aqueous-phase ·OH oxidation, dry and wet deposition.

*Data availability*. All data related to this article are available at https://doi.org/10.7910/DVN/O7HKJQ.

*Author contributions*. JMGS performed all kinetic experiments, treated all data, and built the atmospheric implication
discussion. NB provided the UV-VIS data for $RONO_2$. JMGS and SR developed the UHPLC-UV method for $RONO_2$. JMGS and JLC performed the organic synthesis of $RONO_2$. AM and JLC led the work. JMGS, AM, and JW wrote the article with inputs from all co-authors.

*Competing interests*. The authors declare that they have no conflict of interest.




*Acknowledgements*. The authors thank Camille Mouchel-Vallon for his help using the GECKO-A modelling tool.

*Financial support*. This project has received funding from the European Union's Horizon 2020 research and innovation program under the Marie Skłodowska-Curie (grant no. 713750). It has been carried out with the financial support of the

Regional Council of Provence-Alpes-Côte d'Azur and with the financial support of the A*MIDEX (grant no. ANR-11-IDEX-0001-02), funded by the Investissements d'Avenir project funded by the French Government, managed by the French National Research Agency (ANR). This study also received funding from the French CNRS-LEFE-CHAT (Programme National-Les Enveloppes Fluides et l'Environnement-Chimie Atmosphérique – Project "MULTINITRATES"). The authors also acknowledge the support from the French National Research Agency (ANR-PRCI) through the projects PARAMOUNT

(ANR18-CE92-0038-02) ORACLE (ANR-20-CE93-0008-01_ACT) and AEROFOG (ANR-22-CE92-0051).

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
