# Peer review of "On the importance of multiphase photolysis of organic nitrates on their global atmospheric removal"

_EGUsphere, 2023_

## Author Response (AR1)

The authors thank the reviewers for their positive feedbacks on our work and their helpful comments and questions. We have addressed all comments and corrections as detailed below, and we modified the manuscript accordingly.

**REFEREE #1**

**Specific comment**

**The authors should more thoroughly discuss the differences in the products formed from photolysis, OH reactions, and hydrolysis in the gas and condensed phase and whether they serve as sources of NOx. For example, although OH will react with alkyl nitrates, H atom abstraction at the a-carbon and b-carbon atoms, which will lead to NO2 as a product, will be negligible. Abstraction at other sites will dominate, with the products being multifunctional nitrates rather than those that release NO2. Sufficient information should be available in the literature about the products of these other processes to evaluate their impact on NOx.**

--> This is clearly important. However, up to date, the reaction products of $RONO_2$ reactivity are still unknown in the condensed phases. Moreover, a thorough discussion on the $NO_x$ recycling efficiencies can warrant an article on its own. Currently, we are working on two articles that focus on the fate of $RONO_2$ under aqueous-phase reactivity and the $NO_x$ recycling efficiencies of this reactivity. We plan to include this discussion once the aqueous-phase photoreactivity mechanisms are presented.

In this present article, our goal was to establish the importance of each sink for each class of $RONO_2$ and we briefly discussed the potential ability of each family of $RONO_2$ to transport $NO_x$. For example, alkyl nitrates have long lifetimes and their sinks are dominated by gas phase chemistry, thus they likely play a significant role in the flatter $NO_x$ distribution compared to terpene nitrates, which have shorter lifetimes, and their sinks are controlled by deposition and aqueous phase chemistry.

**Technical Comments**

**1. The x-axis labels in figures 4–6 are pretty awful. I don't know what to recommend, other than perhaps replacing the compound names with symbols or using smaller text, but I suggest the authors think about how they might be labeled in such a way to make them more readable.**

--> We have enhanced the quality of the figures to improve that.

**REFEREE #2**

**Main comments**

**Figure 1. There seems to be a contradiction between the main plot and the inner one. In the latter, the solar irradiance is higher than the lamp around ~330 nm, in contrast with the main plot.**

--> Due to the differences on intensities between the lamp and solar spectra, two axes are used in the figure (left and right, respectively). The difference in scaling between the main and the embedded plot may have created the appearance of a contradiction between the two graphs. To avoid this issue, we have adjusted the figure to maintain a consistent ratio between the axes. Furthermore, the right axis is displayed in the same color as the solar actinic flux trace.

**l. 178 The text mentions 2-nitroxy-1-ethanol, whereas the Figure 3 mentions 2-nitroxy-1-propanol. Roberts and Fajer investigated 1-nitrooxy-2-ethanol.**

--> We have investigated the absorption cross-sections and the photolysis rate constants of 1-nitrooxy-2-propanol. However, since there is no gas phase data for this molecule, Figure 3 shows the gas-phase

absorption cross-section of 1-nitrooxyethanol. We included this due to the high similarity of their chemical structure. To clarify this, we have modified the text as it follows and included both chemical structures in the figure with a color code.

"The absorption cross-sections of 1-nitrooxy-2-propanol have not been investigated in the gas phase. Fig. 3b compares its aqueous-phase absorption cross-sections with those of gas-phase 1-nitrooxyethanol (in red in Fig. 3b), determined by Roberts and Fajer, (1989). The only other β-hydroxy $RONO_2$ for which gas-phase absorption cross-sections were investigated, i.e., *trans*-2-nitrooxy-1-cyclopentanol, is not shown here since the molecule does not absorb UV light above 275 nm (Wängberg et al., 1996)."

**Figure 3. There is something wrong with the gas-phase cross sections of alpha-nitrooxyacetone shown on this figure. From the MPI page on cross sections (http://satellite.mpic.de/spectral_atlas/cross_sections/), the cross section from Roberts & Fajer should be about 4 E-20 cm2 molecule-1 at 300 nm, only slightly below the values from Barnes. Please check. There seems to be also a problem for the cross sections of 1-nitrooxy-2-ethanol: at 290 nm, the values from Roberts & Fajer are close to 4 E-21 cm2 molec-1 at 290 nm.**

--> Yes. It appears that we made a mistake while digitizing the figure that displays the absorption cross-sections of those two molecules. We have accordingly updated the figure and all values calculated from those were corrected.

**l. 235-236: However recent work indicates very fast photolysis of particulate nitrate e.g. in sea salt and in other aerosols (e.g. Andersen et al. 2023; https://www.science.org/doi/10.1126/sciadv.add6266). Clearly nitrate absorption cross sections are greatly enhanced, but also the quantum yield is likely very high.**

--> Yes. Nevertheless, the fast photolysis of nitrate is suspected to occur in an heterogenous phase where the absorption cross-sections are enhanced compared to bulk solutions, but the quantum yields are not lowered. The lines 235-236 referred to nitrate photolysis in bulk solutions. In solution, nitrate quantum yields are much lower. We have included "bulk aqueous solutions" to clarify this point.

**SI Section S5: "suspected to be an impurity": isn't that too strong? The main text suggested that the high values could be real.**

--> Yes. It is difficult to say if it is real or not despite the purification. We have modified the text to "which may be due to an impurity".

**l. 316: At what temperature were the KH calculated? Note that typical temperatures in clouds are lower than room temperature (maybe of the order of 280 K).**

--> Unfortunately, only 298 K kinetic rate constants are available for $RONO_2$. Therefore, all $K_H$ were calculated at 298 K. Furthermore, most of the $RONO_2$ experimental $K_H$ values were investigated at this temperature. We are aware that typical cloud temperatures are lower than that value, and thus we have added the following sentence in the manuscript:

"All $K_H$ values are set at 298 K, as well as all reactivity kinetic rate constants for which most of the activation energies are unknown. Note that lower temperatures should be more realistic, they should mostly affect $K_H$ values, therefore, our results probably underestimate the atmospheric fractioning to the aqueous phase."

**l. 329 A value of 2.9 E-7 is given in the text but I cannot find it in the Table. A vlue of 4.16 E-7 s-1 is given for many compounds. Same for the value of 6.6E-5 s-1 which seems to be 1.09 E-4 s-1 in the Table.**

**l. 335 For the dinitrate, a value of 7.57 E-7 s-1 is found in the Table, in contradiction with the text.**

--> Thanks for identifying those mistakes. The values given in the text ($3.9 \cdot 10^{-7}$, $6.6 \cdot 10^{-5}$, and $4.4 \cdot 10^{-6}$) are the correct ones that were used for the calculations. The values on the Table correspond to old values which were not correctly updated. We have updated the tables showing the correct values.

**Figure 4: I don't know how to interpret this figure. For example, for alpha-nitrooxyacetone, the lifetime shown by the plot is ~250 hours, which is certainly not correct.**

--> The atmospheric lifetime of alpha-nitrooxyacetone was displayed on the right axis (and corresponded to a value of 28 h), the same as for all other isoprene and terpene nitrates. To avoid confusion, we have separated the graph into two panels with different scales. The first panel is for small $RONO_2$ with much longer lifetimes, while the second panel is for the rest of $RONO_2$ with shorter lifetimes.

**l. 427 Insert "daytime averaged" before "RONO2 deposition"**

--> We have included "daytime averaged" to that sentence.

**l. 427: The value of 1.4 cm/s is wildly inapprapriate for monofunctional nitrates, and also too high for the monoterpene nitrates. The study of Russo et al (www.atmos-chem-phys.net/10/1865/2010/) reports a deposition velocity of 0.13 cm/s for methylnitrate. For terpene nitrates, Nguyen et al. report values well below 1 cm/s. I recommend using separate deposition velocities for the different groups of compounds.**

--> Thank you for the suggestion, it has largely improved the accuracy of the calculations. We have assigned a value of:

- 0.15 cm s$^{-1}$ for alkyl nitrates based on Abeleira et al., (2018), who determined the deposition velocities of several alkyl nitrates;

- 0.8 cm s$^{-1}$ for terpenes from Nguyen et al., 2015;

- 1.5 cm s$^{-1}$ for the rest of our $RONO_2$ (polyfunctional $RONO_2$ bearing less than 10 carbon atoms), averaged from deposition velocities determined by Nguyen et al., (2015) for this class of $RONO_2$.

Since the atmospheric lifetimes of alkyl nitrates are much longer than the rest of $RONO_2$, we have split each graph into two panels in the same manner as for Figure 4.

**I find the entire Section 4 interesting but very long. Wouldn't it be possible to shorten it? I think there is sufficient redundancy between the different figures (4, 5, 6) to remove one of them.**

--> Yes, we agree with the reviewer. However, we believe that all three figures provide significant insights. Figure 4 demonstrates the importance of aqueous phase ·OH oxidation compared to aqueous phase photolysis. Figure 5 suggests that hydrolysis may be a major sink for some $RONO_2$, while for others, aqueous phase ·OH oxidation can compete or even exceed hydrolysis as a sink. Finally, Figure 6 provides atmospherically relevant lifetimes, considering deposition sinks. While it would be possible to condensing all figures into one, doing so would make the figure more complex and difficult to read.

To shorten Section 4, we have moved detail on the methodology used to assign the kinetic rate constants to the SI. As a result, Section 4 now focusses on the results of the calculations and should look less redundant.

**Table S7 and Table S9: not clear what is tau_chem. Maybe the tau_chem columns could be dropped, as those Tables are quite large.**

--> Tau_chem represent the chemical multiphase lifetimes (without deposition). We have followed the reviewer's suggestion and dropped them since lifetimes including deposition are more relevant.

**Technical/language comments**

**l. 18 weird sentence. I suggest "... polyfunctional RONO2s, in combination with estimates for the other atmospheric sinks..."**

--> Done.

**l. 22 "can increase by up to 100%"**

--> Done.

**l. 31 "responsible for homogenizing the distribution..."**

--> Done.

**l.32 "In parallel": ??**

--> Changed to "Besides,"

**l. 36 "relative importance as NOx reservoir and sink"**

--> Done.

**l. 37 "the overall rate of transformation"**

--> Done.

**l. 41 "by the RONO2 chemical structure"**

--> Done.

**l. 199 "is the calculated..."**

--> Done.

**l. 212-213 "cannot be generalized to their molecules"**

--> Done.

**l. 221 "that the maximum values (J_calc) were similar..."**

--> Done.

**Figure 3: please also show the structure of the chemical compound in the case of 1-nitrooxy-2-ethanol (or propanol?).**

--> Done.

**SI Section S5: "determined" --> "experimental"**

--> Done.

**l. 242 Müller et al. 2014**

--> Done.

**l. 253 "enhanced in liquid cloud droplets"**

--> Done.

**l. 258 "of several RONO2 compounds"**

--> Done.

**l. 275 "structure inferred quantum yield": weird, not clear**

--> Changed to "Likely, this molecule presents a quantum yield lower than 0.07".

**l. 278 "between 310 and 340 nm"**

--> Done.

**l. 280-281: "since they indicate...."**

--> Done.

**l. 281-292 "irrespective of the functional groups besides the nitrate function"**

--> Done.

**l. 285 "by a factor between 2.4 and 3.5 when using a..."**

--> Done.

**l. 286 "The comparison shows that.... lifetimes of alkyl nitrates are relatively similar."**

--> Done.

**l. 293 "The compared pair": weird**

--> Changed to "the pair of β-hydroxynitrates compared".

**l. 293 Were 1-nitrooxy-2-propanol and 2-nitrooxyethanol both investigated?**

--> Answered above.

**l. 298 "were calculated, as exposed..." (comma)**

--> Done.

**l. 386 "towards": weird. Please re-phrase.**

--> Changed to "To assess the relative importance of aqueous-phase ·OH-oxidation and photolysis in relation to hydrolysis".

**l. 400 delete the parenthesis**

--> Done.

**l. 422-423: "isoprene and terpene"**

--> Done.

**Caption of Table S7 (also Table S9): the LWC is wrong for wet aerosol.**

--> Done.

**Caption of Table S8: it should state that the values are given for gas-phase photolysis.**

--> Done.